# The Mechanism of Land Registration Program on Land Transfer in Rural China: Considering the Effects of Livelihood Security and Agricultural Management Incentives

**Lei Xu [1,2]**, **Shixiang Chen [1,2,\*]** and **Shuliu Tian [1]**

[1]  School of Political Science and Public Administration, Wuhan University, Wuhan 430072, China
[2]  Local Government Public Service Innovation Research Center, Wuhan University, Wuhan 430072, China
\*  Correspondence: chensx@whu.edu.cn; Tel.: +86-027-6875-5689

**Abstract:** The key to a smooth land transfer (including land transfer-out and transfer-in) lies in the cooperation between the land supply and demand parties. Existing studies explore how land registration programs affect land transfer from a macro level or from a micro level in a certain area, but little consideration has been given to the interaction and behavioral disciplines of stakeholders. This article aims at testing the possible mechanism of the land registration program on land transfer in rural China by bridging and extending concepts from peasant theories and by employing mediation models. The empirical results reveal that the land registration program has a significant positive impact on land transfer, which is an important path in order to overcome the cooperative dilemma between land supply and demand parties. Additionally, livelihood security inhibits the positive impact of the land registration program on land transfer-out. While agricultural management incentives promote the positive impact of the land registration program on land transfer-in. Furthermore, these findings contribute to a novel perspective for evaluating land registration programs and deepen the understanding that intricate driving factors behind the decrease in the land transfer growth rate can have in rural China.

**Keywords:** land registration program; land transfer; behavioral disciplines; livelihood security; agricultural management incentives; China

## 1. Introduction

With the implementation of China's Rural Revitalization Strategy, the small, scattered, and weak land-use problem has gradually become a major restriction to China's rural economic development [1,2]. A moderate scale of land transfer is not only conducive to promoting the integration of land resources but also directly affects the overall development level of agricultural modernization [3]. The Chinese government has issued a series of policies to encourage land transfer to a certain extent [4,5]. Among them, land registration programs have played extremely important roles.

According to the Statistical Annual Reports of China's Rural Management (excluding Tibet, Taiwan, Hongkong, and Macao), from 2017 to 2020, the national average area of land transfer reached 34.13 million ha (512 million mu), 35.93 million ha (539 million mu), 36.93 million ha (554 million mu), and 35.47 million ha (532 million mu), respectively. On the provincial scale, an upward trend of land transfer proportion from 2005 to 2020 is shown in Figure 1. In 2005, the land transfer rate in 26 provinces did not exceed 10%. In 2008 and 2011, only the land transfer rate in Shanghai exceeded 50%. Until 2020, the land transfer rate in Beijing, Shanghai, Jiangsu, Zhejiang, and Heilongjiang has exceeded 50%. Meanwhile, the land transfer rate in other provinces has increased. Especially, from 2005 to 2020, the land transfer rates of Anhui, Tianjin, and Shandong have increased 42.84%, 40.89%, and 40.85%, respectively. However, the growth rate of land transfer in the country as a whole has gradually slowed down. For example, from 2011 to 2014, the year-on-year

growth rates of land transfer were 3.19%, 3.40%, 4.46%, and 4.66%; from 2015 to 2017, the rates dropped to 2.94%, 1.70%, and 1.90%, respectively (Statistics from the Ministry of Agriculture and Rural Affairs of the People's Republic of China). As mentioned above, it seems that land registration programs have not achieved the expected effects of the policies.

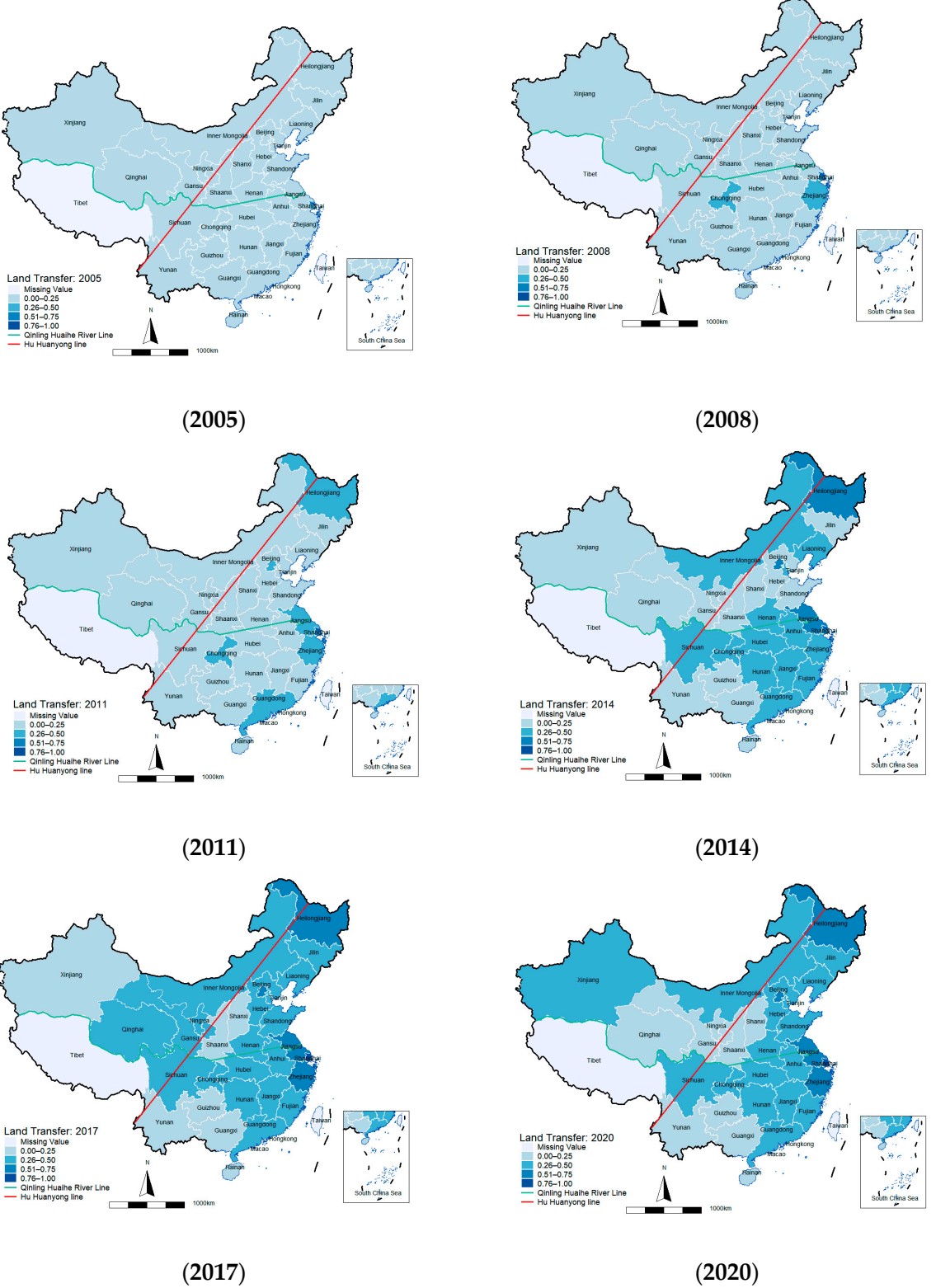

**Figure 1.** Land transfer proportion in rural China from 2005 to 2020.

In terms of studies about the impacts of land registration programs on land transfer, some scholars support the notion that land registration programs could increase the land transfer rate. For example, the theoretical study by Heltberg [6] and the empirical studies by Jin et al. [7], Min et al. [8], Macours et al. [9], and Yami et al. [10] report that unclear land property rights and frequent land adjustment hinder land transfer and reduce the efficiency of agricultural production. As a mandatory system change model, land registration programs provide a set of legalized rules and procedures for land transfer, reconstruct the relationship of land property rights by legal empowerment, protect the economic benefits of the main stakeholders, and then improve tenure stability and transferability of land [11–13]. However, other scholars hold the opposite views. For example, Place et al. [14] believe that there is no causal relationship between land registration programs and land transfers, and to some extent, they may even lead to land conflicts. First, Deininger et al. [15] and Hombrados et al. [16] propose that land registration programs strengthen farmers endowment effect on land, which means the increase of the land supply party exclusivity and land transaction costs. Second, Besley et al. [17] report that land registration programs curb the investment effect of land due to the small size of farmers' land and the constraints of the rural financial market. In addition, the empirical study by Gould [18] reveals that the top-down land registration programs may deviate from the reality of rural society. Holden et al. [11], Saint-Macary et al. [19], and Toulmin [20] also point out that land registration programs violate the informal land tenure system and cause new conflicts, which is not conducive to the protection of farmers' interests. In summary, there is no unified understanding about the impacts of land registration programs on land transfer in the academic circle.

Although existing studies explore how land registration programs affect land transfer [21–25], very few involve the cooperation and behavioral disciplines of stakeholders in the process of land transfer. In China, land rights are divided into ownership, contracting rights, and management rights. Land transfer is defined as the practice that farmers retain the contracting rights and transfer the management rights only to other farmers or economic organizations by subcontracting, transferring, exchanging, cooperating, investing, leasing, and mortgaging, without changing its agricultural use [26]. With the promotion and advocacy of the land registration program, the land transfer transaction requires cooperation between the land supply and demand parties. In addition, the two main parties often follow a certain discipline to make corresponding transfer decisions. As a result, evaluating the mechanism of the land registration program on land transfer (including land transfer-out and transfer-in) from the perspectives of behavioral disciplines, based on mediation models and CHFS 2015, is of great significance.

The following sections are as follows: Section 2 presents the institutional background, theoretical construction, and research hypotheses; Section 3 describes the empirical research design, including data source, variable definition, and model setting; Section 4 presents the descriptive and empirical results; Section 5 discusses them; and Section 6 presents the conclusions and implications.

## 2. Institutional Background and Theoretical Analysis

### 2.1. Institutional Background

In order to improve tenure stability and transferability of land, the Chinese government vigorously implemented a land registration program. In general, China's 70 year evolution of the rural land registration program can be divided into two stages (Figure 2).

In the first stage, the Land Reform Law of the People's Republic of China in 1950 completely abolished the feudal land system. To guarantee that every cultivator has their own land, the country implemented the peasant land ownership system, and the local government was responsible for issuing land and house ownership certificates. In 1978, under the Household Contract Responsibility System, the collective land in rural areas was distributed equally by per capita, which mobilized farmers' enthusiasm for agricultural production. However, based on the principle that only when the number of people increase, land can be increased and vice versa, irregular land adjustment by rural

collectives occurred from time to time, and the Household Contract Responsibility System did not endow farmers with long-term land contracting and management rights [27–29]. In order to stabilize the property rights of land and increase the enthusiasm for land investment, the No.1 Document in 1984 did endow farmers with 15 year land contracting and management rights, which was extended to 30 years in 1993.

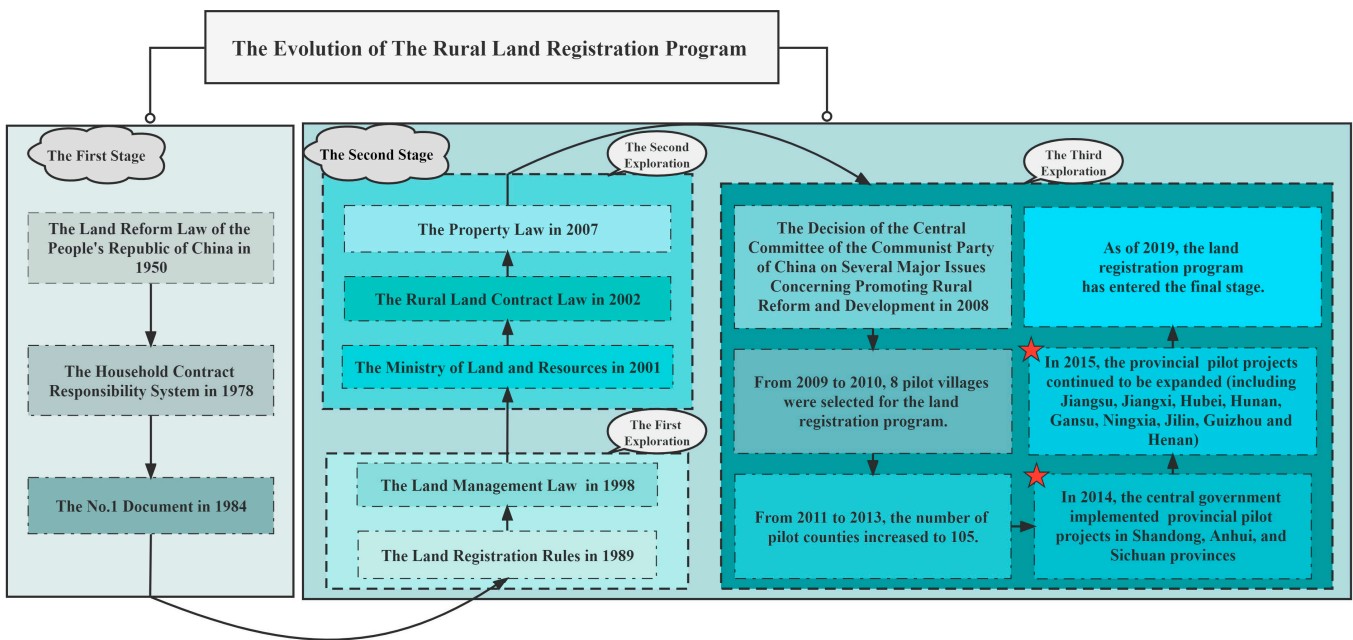

**Figure 2.** The evolution of the rural land registration program.

Soon thereafter, the Chinese government started the second stage of the exploration of the land registration program, which mainly consisted of the following three attempts:

1. For the first time, the Land Registration Rules in 1989 started the general land registration, and clearly stipulated that the acquisition, change, or transfer of land is invalid without registration. The Land Management Law in 1998 extended a second round of land contracting and management rights to 30 years and enforced the distribution of land registration and certification by law for the first time;

2. The second time was the Ministry of Land and Resources in 2001. In addition, the Rural Land Contract Law in 2002 clearly required rural collective economic organizations or villagers' committees to issue a formal land contract and a certificate of land contracting and management rights to farmers. The Property Law in 2007 exactly defined land contracting and management rights as usufructuary rights. The land certificate was the basis for the mortgage of management rights;

3. For the third time, the Decision of the Central Committee of the Communist Party of China on Several Major Issues Concerning Promoting Rural Reform and Development in 2008 clearly gave farmers more guaranteed land contracting and management rights. The existing land contracting relationship should be kept stable and unchanged for a long time, and the contracted land should be confirmed, registered, and certified, which fully reflects the stability and continuity of contracting rights. This round of the land registration program is gradually promoted, from the initial pilot work within the villages, townships, counties, and provinces, and then extended to the whole country. Specifically, from 2009 to 2010, it was determined that eight villages in eight provinces, including Sichuan Province, took the lead in registering land contracting and management rights and issuing certificates to households; from 2011 to 2013, pilot projects were carried out in hundreds of counties. Among them, in 2011, the pilot program for land registration was expanded to 50 counties, covering 710 townships and 12,150 villages in 28 provinces (autonomous regions and municipalities). In 2013,

105 counties (cities, districts) were further identified as pilot areas for the registration of land contracting and management rights nationwide; in 2014, the central government implemented provincial pilot projects in Shandong, Anhui, and Sichuan provinces; in 2015, the pilot projects continued to be expanded and nine provinces (autonomous regions), including Jiangsu, Jiangxi, Hubei, Hunan, Gansu, Ningxia, Jilin, Guizhou, and Henan, were selected to carry out pilot projects in the whole province. As of 2019, the registration and certification of rural contracted land had entered the final stage.

Thus, in order to explore the mechanism of the land registration program on land transfer in rural China, it is necessary to develop research focusing on this critical period of policy implementation and promotion at a provincial level. As such, considering the availability and validity of data, this article selects CHFS 2015.

### 2.2. Theoretical Analysis

2.2.1. Land Registration Program: Overcome the Cooperative Dilemma between Land Supply and Demand Parties

Based on the direction of land transfer, land transfer behavior, as theorized by Yang et al. [26], can be classified into two categories: land transfer-out and land transfer-in. Correspondingly, the main stakeholders are the land supply and demand parties. More specifically, the land supply party refers to farmers who are willing to transfer out all or part of the land management rights. The land demand party refers to individual farmers or agricultural economic organizations that have an objective demand for moderately scaled land, such as large farmers, leading enterprises, agricultural companies, or professional farmer cooperatives.

Scholars have not yet reached a consensus on the impacts of land registration programs on land transfer. In discussing determinants of the main stakeholders' behavioral decisions (including land supply and demand parties), the mainstream views are that land registration programs improve the land rental market and then promote land transfer for three main reasons [21,30–32]: (1) Widening options of tenants. Households could transfer their lands to higher-efficiency farmers or agricultural organizations in long-term contracts and get a more satisfactory rental price; (2) broaden households' income sources. Those farmers who remain in agricultural operations can transfer in moderately large-scale land and obtain more productive income and agricultural subsidies; (3) developing favorable institutional conditions of land registration programs, such as the stability of property rights, preferences for land management rights mortgage loans, or reduction of land transaction costs. Accordingly, most scholars conclude that land registration programs encourage the land demand party to increase agricultural investment and transfer more land [33,34]. As for the land supply party, related studies propose that, on the premise that farmers' land contracting and management rights are fully protected, they act in accordance with the principle of profit maximization and tend to transfer their land out for higher economic benefits [35,36].

Under Chinese government policy, land transfer should adhere to the principle of voluntariness. More importantly, a top priority is given to getting farmers' permission [37]. In 2014, the General Office of the Central Committee of the Communist Party of China and the General Office of the State Council issued the Opinions on Guiding the Orderly Circulation of Rural Land Management Rights to Develop Agricultural Appropriate Scale Operations, outlining the provisions that farmers could decide on the form and price of land transfer. In such a policy background, some scholars assert that as for the restraints of land transfer programs on the subject, scope, and content of land property rights, the land demand party should follow and respect the land supply party's willingness to transfer land [30,31]. Therefore, the land supply and demand parties are closely interdependent. The land transfer transaction can go smoothly only when the two parties reach an agreement. Cooperation between transfer-in and transfer-out means that, through communication and negotiation, the land supply party and the land demand party reach a consensus on the transfer method, transfer period, rent, etc. When farmers are willing to transfer their land

out, other farmers or agricultural organizations are willing to transfer this land in. Based on this, this article proposes hypothesis 1.

**Hypothesis 1.** *The land registration program has a significant positive impact on land transfer (including land transfer-out and transfer-in).*

2.2.2. Land Registration Program, Behavioral Disciplines, and Land Transfer

Behavioral disciplines reflect individuals' reasons for action. Those are set by the individuals according to the situation and dominant needs of themselves and their families. In the scenario of land transfer, stakeholders (including the land supply and demand parties) following a specific discipline take full advantage of land resources, and promote the value attached to land on the premise of maximizing family benefits and minimizing risks [38,39]. That is, whether households make land transfer decisions depends on their judgment of the expected value of land. As theorized by Krutilla [40] and Bishop [41], land value can be classified into two categories: market value (it can be directly converted into currency through market transactions) and non-market value (it cannot be realized through market transactions but objectively exists). Consistent with the classification of land value, the behavioral disciplines of households can be divided into livelihood security and agricultural management incentives, as shown in Table 1.

**Table 1.** Behavioral disciplines, land value, and forms of behavior.

| Behavioral Disciplines | Land Value | Land Value Classifications | Forms of Behavior |
|---|---|---|---|
| Livelihood security | Non-market value | Social security value | Increasing households' living security and repelling the land transfer-out, specifically, land bears the dual responsibility of endowment insurance and employment insurance, and has a strong social security function. In the context of the imperfect rural social security system, farmers lacking livelihood security refuse to transfer their land out. |
| Agricultural management incentives | Market value | Land output value | Increasing the incentives for agricultural operations and transferring in moderately large-scale land, specifically, adopting more modern agricultural machinery and technology; investing in more agricultural materials and products; spending more time on agricultural production and operation; increasing the output value and adding value of agricultural products; and obtaining more productive income from farmland management. |
| | | Land rental value | Increasing land transfer rents, specifically, negotiating with agricultural business entities through more effective ways to obtain higher property income from land transfer. |
| | | Agricultural subsidies | Establishing good relations with local governments, responding to problems in a timely manner, and obtaining agricultural subsidy income. |

1. Land Registration Program, Livelihood Security, and the Behavior of Land Supply Party

As the smallest economic unit in agricultural production, factors such as rural labor migration and land use structure adjustment (including land transfer) are mirrored in the rational plans of households based on the utility maximization principle [42,43]. In fact, increasing the efficiency of productive factors and allocating resources does not occur independently [44]. Some researchers have concluded that rural labor migration could significantly affect farmers' land transfer decisions. Moreover, the higher the rural labor force's off-farm employment rate, the higher the incidence of land transfer [45–47]. However, for China, a country with a traditional agricultural culture that has lasted for thousands of years, farmers' persistence in the land right has not substantially changed. Due to the increasing proportions of rural part-time farmers, agricultural machinery inputs and agricultural feminization [48], resource endowment [49], farmer risk consciousness [50], and the social security function of rural land [51,52], off-farm labor migrants often do not transfer their land out. In particular, in economically underdeveloped areas, households are deeply dependent on their lands for livelihood security, which is consistent with Scott's survival ethics theory. Livelihood security depends on the sensitivity and resilience of farmers to risk disturbances inside and outside their families. The higher the level of risk perception and resistance of farmers, the safer the living conditions of their families, and vice versa. In addition, suppliers of the land transfer market always follow the safety-first principle [53]. That is, farmers prefer to minimize the probability of having a disaster rather than maximize their average return when they are at risk of any major loss that could endanger their subsistence. In some rural societies where the social security system still needs to be improved, reserving land is the last resort for farmers. The Chinese government has implemented a series of land tenure reforms to improve tenure stability [54]. Under the land registration program, lessees can be endowed with assurance of land management rights [55]. It is worth noting that there are still problems such as inadequate land measuring, historical remaining problems, and difficulty in handling conflicts and disputes, resulting in a lag in the progress of the land registration program. Although farmers can obtain more rental value through land transfer, in practice, some farmers still refuse to transfer their land out to preserve the non-market value of land, namely, social security value. In other words, for those farmers following the discipline of livelihood security promotion, the operation of the land transfer-out is against their behavioral logic.

2. Land Registration Program, Agricultural Management Incentives, and the Behavior of Land Demand Party

Agricultural management incentives refer to increasing economic entities' incentives for agricultural operations, specifically, adopting more modern agricultural machinery and technology, investing in more agricultural materials and products, and spending more on agricultural production and operation time in order to increase the output value and adding value of agricultural products and obtain more productive income from farmland management. A strengthening of land rights security promotes households' investment willingness and capacities through three main channels [33,56,57]: (1) Due to a reduction of administrative intervention and expropriation risks, farmers could anticipate the secure and expected returns; (2) because of the availability of land management rights as collateral, the possession of formally recognized land contracting and management rights may be a signal of lower credit risk and can improve households' access to formal credit; and (3) the certification and registration of land management rights reduce transaction costs. Furthermore, the increasing flexibility in returns encourages households' investment incentives in longer-term, land-saving investments. This is because, under the background of the land registration program in rural China, land management rights clarification is expected to reduce the risk associated with making long-term agricultural investments. This, in turn, should increase demand for land transfer [58,59]. Specifically, if the land demand party is an agricultural economic organization such as leading enterprises or professional farmers' cooperatives, then seeking the maximization of economic benefits may be the behavioral

rule followed by them. They incline to transfer in moderately large-scale land for more agricultural production and management income [60–63]. The land demand party is not only limited to agricultural economic organizations but also includes individual farmers. Based on the rational peasant theory, farmers have a sensitive response to price and other market competition, and have no difference from other investors [64,65]. Specifically, farmers who act on the principle of safety first do not give up all opportunities to pursue benefits but set up a subsistence crisis level based on the actual situation of the households. The subsistence crisis level, perhaps a "danger zone" rather than a "level" would be more accurate, is a threshold below which the deterioration in subsistence, security, status, and family social cohesion is massive and painful. For those near-subsistence farmers, risk aversion may be quite strong because the returns above expected values may not offset the severe penalties for returns below the expected values; above which the discipline of seeking to maximize profits prevails. And farmers are rational in the process of agricultural investing [53].

Thus, in order to optimize land resources and maintain the value attached to land, the main stakeholders (including the land supply and demand parties) in the process of land transfer always make relevant judgments and decisions based on a certain behavioral discipline. The land registration program affects the stakeholders' behavioral disciplines, which in turn determines the actual incidence of land transfer. Accordingly, this article proposes Hypothesis 2a and Hypothesis 2b.

**Hypothesis 2a.** *For the land supply party, livelihood security inhibits the positive impact of the land registration program on land transfer-out.*

**Hypothesis 2b.** *For the land demand party, agricultural management incentives increase the positive impact of the land registration program on land transfer-in.*

Based on these three hypotheses, a theoretical analysis framework for this article is shown in Figure 3.

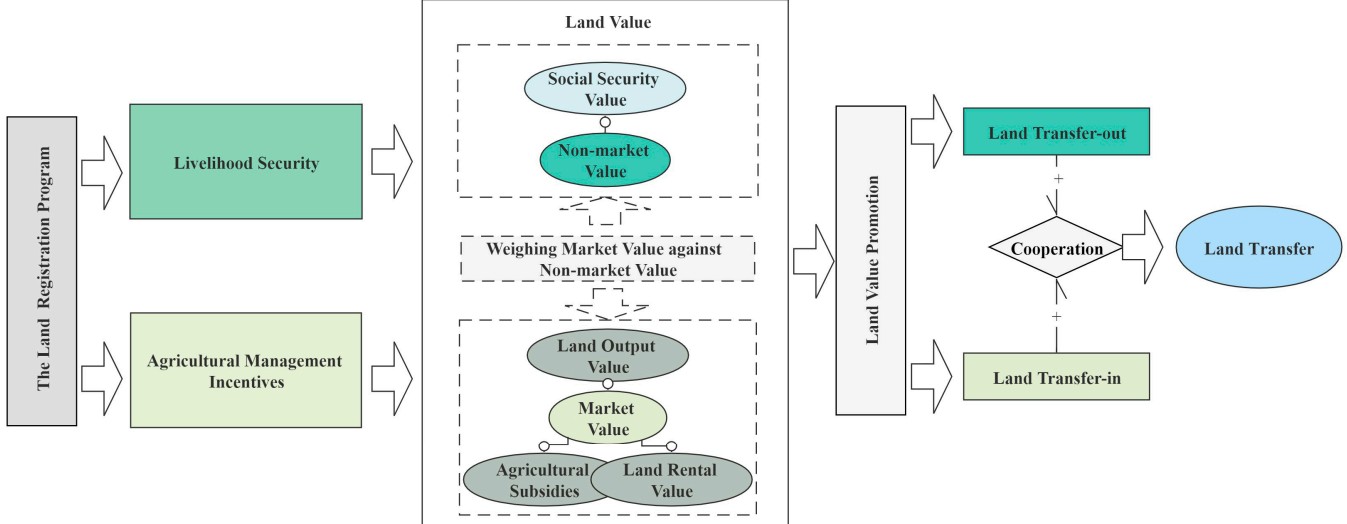

**Figure 3.** Theoretical analysis of the land registration program, behavioral disciplines, and land transfer.

## 3. Data Source, Variable Definition, and Empirical Approach

### 3.1. Data Source

This article uses the representative micro-data from the China Household Finance Survey (CHFS), organized and managed by the Survey and Research Center for China Household Finance of Southwestern University of Finance and Economics in China [66]. Given the evolution of the rural land registration program, it is necessary to establish the mechanism of the land registration program on land transfer during the critical period

in which provincial pilot projects began to be implemented in 2014 (including Shandong, Anhui, and Sichuan provinces) and continued to be expanded in 2015 (including Jiangsu, Jiangxi, Hu-bei, Hunan, Gansu, Ningxia, Jilin, Guizhou, and Henan provinces). Furthermore, the latest data released by the center does not contain related items. We selected the survey conducted in 2015 for analysis. To ensure the representativeness of samples, the team conducted scientific investigations in 29 provinces, 351 counties, and 1396 villages across the country, and the sample size of CHFS 2015 reached 37,289 households. The survey has refined the relevant indicators of rural household economics and farmers' financial behavior and provided high-quality micro-data on the issue of land ownership in China.

In order to improve the validity of the data for this research, we screened samples according to the following criteria: (1) Using STATA14SE software to match cities, households and personal data from CHFS 2015; (2) only selecting rural households as samples; (3) deleting samples with missing and abnormal key variables; (4) conducting bilateral tail-shrinking treatment for continuous variables based on the 1% standard; and (5) taking the logarithm of continuous variables to control the interference of extreme values on the conclusion. After data cleaning, the capacity of a valid sample is 17,310 (each household sample contains more than one individual sample). The valid sample size is 17,310 individual samples of residents.

### 3.2. Variable Definition

3.2.1. Explained Variables

1. According to the direction, this article classifies land transfer behavior into two categories: land transfer-out and transfer-in. In the original questionnaire, land transfer-out is measured by the question "C5005b: Have you transferred the land's right of management to another person or entity?". Land transfer-in is measured by the question "C5011a Does your household have inward transferred land?". Furthermore, land transfer is measured by the combination of C5005b and C5011a. Therefore, the explained variables are binary. More specifically, 1 if a household has transferred land, transferred land out, or transferred land in, or 0 otherwise.

3.2.2. Explanatory Variable

2. The implementation of the land registration program has become a key stage for China to accelerate from traditional agriculture to modern agriculture. In practice, there are great differences in the progress of land contracting and management rights registration and certification among households in rural villages. Based on this, the explanatory variable-land registration in this article is set by the standard that the farmer households actually obtain the land certificate and measured by the question "C5004c Did your household have the land management rights certificate for?" in the questionnaire. Therefore, the explanatory variable is binary. Specifically, 1 if a household has officially registered the land contracting and management rights, obtaining certificates for the confirmation of the land rights, or 0 otherwise.

3.2.3. Mediating Variables

3. Mediating variables are security, job-number, agri-time, crop-yield, and crop-value. Referring to the study of Knutsson et al. [67], two indexes measure livelihood security: security and job-number. More specifically, in the original questionnaire, security is measured by the question "A4027b Do you feel secure living in today's era?" Specifically, 1 if a household feels secure, or 0 otherwise; job-number is measured by the question "A3002 How many jobs do you presently have?", and the answer ranges from 1 to 5. As for agricultural management incentives, this article uses agri-time, crop-yield, and crop-value as the evaluation indexes to measure. More specifically, agri-time is measured by the question "B1003 Last year, for how many months were your family members doing agricultural business? (Unit: month)", and the answer ranges from 1–12; crop-yield is measured by the question "B1004f Last year, the total

crop output of your household was (Unit: kilograms)", and the answer ranges from 0–1720000; crop value is measured by the question "B1004h: Last year, the total crop output of your household was (Unit:yuan)", and the answer ranges from 0–8,000,000.

### 3.2.4. Instrumental Variables

4. Based on the study of Angrist et al. [68], concern and investment-choice are selected as the IVs for analysis. More specifically, concern is measured by the question "A4002a What is your degree of concern for economic and financial information?", and the answer is recorded from 1 (not at all) to 5 (extremely concerned); investment choice is measured by the question "A4003 Which of the choices below do you want to invest in most if you have adequate money?", and the answer is recorded from 1 (unwilling to carry any risk) to 5 (project with high-risk and high-return).

### 3.2.5. Control Variables

5. To eliminate the interference of other factors, based on existing studies by Di Falco et al. [12], Li et al. [13], Besley [33], and Wang et al. [34], control variables selected in this article include gender, health, agri-member, impoverishment, cadre, income, land-acreage, quality, and age. The definitions and coding of all variables are presented in Table 2.

**Table 2.** Variables definition and coding.

| Variable Classes | Variable Name | Variable Definition and Coding |
|---|---|---|
| Explained variables | Land transfer | Whether to transfer land in or out: 1 = yes; 0 = no |
| | Land transfer-out | Whether to transfer land out: 1 = yes; 0 = no |
| | Land transfer-in | Whether to transfer land in: 1 = yes; 0 = no |
| Explanatory variable | Land registration | Whether to officially register the land contracting and management rights: 1 = yes; 0 = no |
| Mediating variables | Security | Whether to feel secure or not: 1 = yes; 0 = no |
| | Job-number | Numbers of household head jobs, job |
| | Agri-time | Months of agricultural business, month |
| | Crop-yield | Yield of crop, kilograms |
| | Crop-value | Value of crop, yuan |
| Instrumental variables | Concern | Level of concern for economic and financial information: 1 = not at all; 2 = seldom concerned; 3 = generally concerned; 4 = very concerned; 5 = extremely concerned |
| | Investment-choice | Level of risk of the investment choices: 1 = unwilling to carry any risk; 2 = slight risk and return; 3 = average risk and return; 4 = slightly high-risk and slightly high-return; 5 = high-risk and high-return |
| Control variables | Gender | Gender of household head: 1 = male; 0 = female |
| | Health | Level of health: 1 = very bad; 2 = bad; 3 = ordinary; 4 = good; 5 = very good |
| | Agri-member | Numbers of family members participating in agricultural production or operation, people |
| | Impoverishment | Whether family is impoverished household: 1 = yes; 0 = no |
| | Cadre | Whether there is a family member working as village cadre: 1 = yes; 0 = no |
| | Income | Numbers of total income, yuan |
| | Land-acreage | Area of household's largest piece of cultivated land, mu |
| | Quality | Level of the quality of household's cultivated land in the contract: 1 = very bad; 2 = inferior; 3 = ordinary; 4 = better than average; 5 = very good |
| | Age | Age of household head, year |

### 3.3. Mediation Methods

If both the regression coefficients c and a of $X_1$ and the regression coefficient b of $X_2$ are significant, there is a mediating effect. However, if the coefficient c' of $X_1$ is significant and the sign of a·b is the same to the sign of c', it belongs to a partial mediation effect; if the signs of the two are opposite, it belongs to a suppression effect as shown in Figure 4 [69,70].

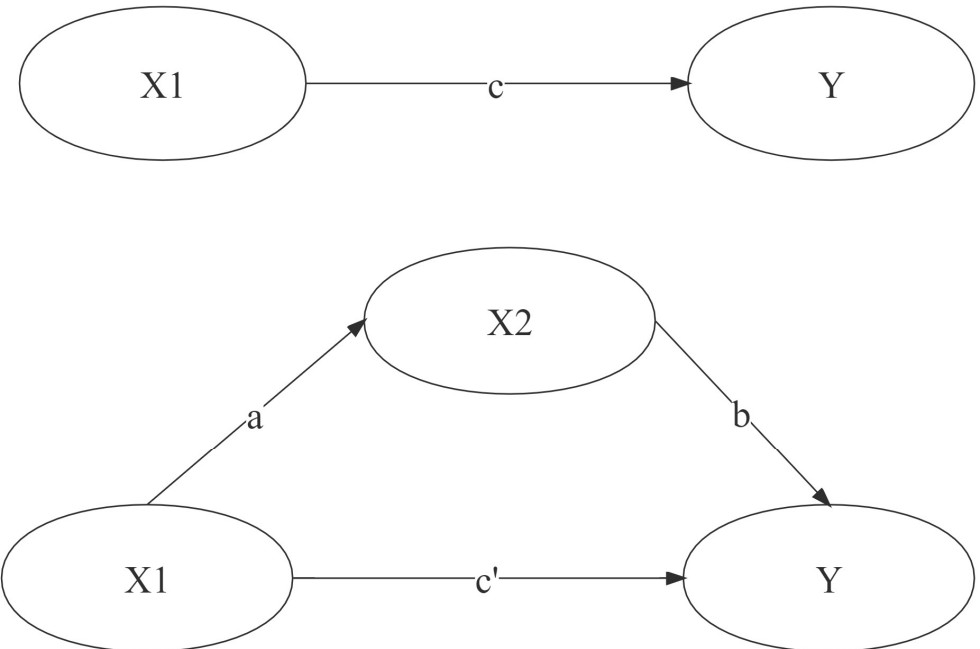

**Figure 4.** Mediation and suppression effects.

A mediation effect is frequently referred to as an indirect effect, where the effect of the explanatory variable $X_1$ on the explained variable Y goes through a mediator $X_2$. The mediation effect is commonly defined as the reduction in the regression coefficient of $X_1$ on Y when the effect of $X_2$ is controlled for. A suppressor is defined as a third variable that increases the regression coefficient between the explanatory variable and the explained variable by its inclusion in a regression equation. When the suppression effect is not controlled for, the relationship between $X_1$ and Y would appear to be smaller or even have the opposite sign.

The main reasons why this article employs the method for estimating the mechanism of the land registration program on land transfer in the binary Logit model are as follows:

1. The explained variables (land transfer, land transfer-out, and land transfer-in) are all binary discrete variables with values of 0 or 1. Due to the fact that the Logit or Probit model has been widely used for households' land transfer behavior analyses [26,44,71];
2. Referring to the study of Breen et al. [72], the method for estimating mediating effects in a Logit or Probit model always performs better than other methods by identifying total, direct, and indirect effects under the sequential ignorability assumption;
3. In order to clearly communicate information and correctly interpret the results from models with binary {0,1} explained variables in a meaningful way, researchers often report marginal effects or odds ratios. Among them, odds ratios are unique to Logit models. Whether to use odds ratios depends on the model specification. If the explanatory variables of interest are discrete, odds ratios are generally preferable to average marginal effects [73].

The basic model is set as shown in Equation (1):

$$y_i^* = x_i' \beta + \varepsilon_i; \ (i = 1, \ldots, n) \tag{1}$$

where $y_i^*$ is a dummy explained variable with a value of 1 or 0, which can be expressed as a latent variable of the respondent's land transfer, land transfer-out and land transfer-in; $x_i'$ represents the respondent's land registration related explanatory variable; $\beta$ is the regression coefficient of the influencing factors; $\varepsilon_i$ is the stochastic disturbance term.

The probability that the respondent makes a potential land transfer/transfer-out/transfer-in is:

$$p\,(y = 1|x) = F\,(x,\ \beta) = \Lambda\,(x'\beta) \equiv \frac{\exp(x'\beta)}{1 + \exp(x'\beta)} \tag{2}$$

Specifically, in order to clarify the quantitative relationship between land registration and land transfer/transfer-out/transfer-in, this article presents benchmark regression analysis and constructs the following three models as shown in Equations (3)–(5).

$$LT_i^* = \beta_0 LRE_i + \Upsilon_0 Z_i + \varepsilon_i;\ \varepsilon_i \sim \text{Normal}\,(0,1) \tag{3}$$

$$LO_i^* = \beta_1 LRE_i + \Upsilon_1 Z_i + \zeta_i;\ \zeta_i \sim \text{Normal}\,(0,1) \tag{4}$$

$$LI_i^* = \beta_2 LRE_i + \Upsilon_2 Z_i + \eta_i;\ \eta_i \sim \text{Normal}\,(0,1) \tag{5}$$

where $LT_i^*$, $LO_i^*$, and $LI_i^*$ respectively represent the latent variables of land transfer, land transfer-out and land transfer-in of respondent i; $LRE_i$ represents whether respondent i has officially registered the land contracting and management rights; $Z_i$ represents other control variables including gender, health, agri member, impoverishment, cadre, income, land acreage, quality, age; $\varepsilon_i$, $\zeta_i$, and $\eta_i$ are stochastic disturbance terms; and $\Upsilon_0$, $\Upsilon_1$, and $\Upsilon_2$ are vectors of parameter estimates. The coefficient $\beta_0$ of Equation (3) is the effect of land registration on land transfer. The coefficient $\beta_1$ of Equation (4) is the effect of land registration on land transfer-out. The coefficient $\beta_2$ of Equation (5) is the effect of land registration on land transfer-in.

In order to exactly define the roles of mediating variables (behavioral disciplines: livelihood security and agricultural management incentives) in the relationship between explanatory variable (land registration) on explained variables (land transfer-in/transfer-out). According to the study of Cheung et al. [69], this article develops mediation models and constructs two sets of recursive equations: Equations (6)–(8) and Equations (9)–(11) to test whether H2a, that is, whether the land registration program affects the behavioral decision-making of the land supply party through livelihood security (LS), and H2b, that is, whether the land registration program affects the behavioral decision-making of the land demand party through agricultural management incentives (AI).

$$LO_i^* = \alpha_0 LRE_i + \lambda_0 Z_i + \delta_i \tag{6}$$

$$LS_i = \alpha_1 LRE_i + \lambda_1 Z_i + \tau_i \tag{7}$$

$$LO_i^* = \alpha_2 LRE_i + \alpha_3 LS_i + \lambda_2 Z_i + \Psi_i \tag{8}$$

where $LO_i^*$ represents the latent variable of the land transfer-out of the respondent i; $LRE_i$ represents whether respondent i has officially registered the land contracting and management rights; $LS_i$ is the suppressor, representing livelihood security (including security and job-number) of respondent i; $Z_i$ represents other control variables including gender, health, agri-member, impoverishment, cadre, income, land-acreage, quality, age; $\lambda_0$, $\lambda_1$, and $\lambda_2$ are vectors of parameter estimates; and $\delta_i$, $\tau_i$, and $\Psi_i$ are stochastic disturbance terms. The coefficient $\alpha_0$ of Equation (6) is the total effect of land registration on land transfer-out. The coefficient $\alpha_1$ of Equation (7) is the effect of land registration on the suppressor-livelihood security. The coefficient $\alpha_2$ of Equation (8) is the direct effect of land registration on land transfer-out after controlling the effect of the suppressor. The coefficient $\alpha_3$ is the effect of the suppressor on land transfer-out after controlling the effect of land registration.

$$LI_i^* = \theta_0 LRE_i + K_0 Z_i + u_i \tag{9}$$

$$AI_i = \theta_1 LRE_i + K_1 Z_i + \mathring{m}_i \tag{10}$$

$$LI_i^* = \theta_2 LRE_i + \theta_3 AI_i + K_2 Z_i + \mathring{q}_i \tag{11}$$

where $LI_i^*$ represents the latent variable of the land transfer-in of the respondent i; $LRE_i$ represents whether respondent i has officially registered the land contracting and management rights; $AI_i$ is the mediator, representing agricultural management incentives (including agri-time, crop-yield, and crop-value) of the respondent i; $Z_i$ represents other control variables including gender, health, agri-member, impoverishment, cadre, income, land-acreage, quality, age; $K_0$, $K_1$, and $K_2$ are vectors of parameter estimates; and $\mathring{u}_i$, $\mathring{m}_i$, and $\mathring{q}_i$ are stochastic disturbance terms. The coefficient $\theta_0$ of Equation (9) is the total effect of land registration on land transfer-in. The coefficient $\theta_1$ of Equation (10) is the effect of land registration on the mediator-agricultural management incentives. The coefficient $\theta_2$ of Equation (11) is the direct effect of land registration on land transfer-in after controlling the effect of the mediator. The coefficient $\theta_3$ is the effect of the mediator on land transfer-in after controlling the effect of land registration.

Furthermore, adequacy of the model, multi-collinearity, heteroscedasticity, autocorrelation, normality, linearity, endogeneity tests are carried out, and diagnostic tests and rectifications are as follows:

1. Adequacy of the model. Goodness of fit is an important basis for evaluating the rationality of the model. The goodness of fit of a logit model can be represented by the percentage of correct predictions [74]. The percentages of correct predictions of households' land transfer, land transfer-out and transfer-in models as shown in Equations (3)–(5) are as high as 73.76%, 93.76%, and 79.46%, respectively. In general, the model set in this article is relatively reasonable;

2. Multi-collinearity. Multi-collinearity problem creates large variance associated with the parameter estimates [75]. This article tests the multi-collinearity of all explanatory variables and shows that the mean VIF is 1.05. This indicates that there is no multi-collinearity problem;

3. Heteroskedasticity and Spatial Autocorrelation. Even though the country pilots the implementation of land registration step by step, from villages, towns, districts, counties, and provinces to the whole country, it always takes the village as a basic unit, and the whole village is implemented. This means that there might be problems with heteroscedasticity and spatial autocorrelation. On the one hand, the model has a heteroscedasticity problem, which is further investigated by the BP test (Chi(2) = 95.09; $p < 0.01$) [76] and the White-test (Chi(2) = 400.49; $p < 0.01$) [77]; on the other hand, even though we use cross-sectional data to analyze (in the case of cross-sectional data, time-series autocorrelation could be avoided), spillover effects-spatial autocorrelation may interfere with the results. We examine the spatial autocorrelation using Moran's I index [78]. The Moran's I values range from $-1$ to $+1$. The higher the absolute values, the stronger the spatial autocorrelation. The statistics tools in GeoDa software could offer Moran's I value along with a $p$-value. More specifically, based on Queen Contiguity, spatial weight matrix is generated. In addition, with random spatial distribution (199 permutations), the results show that Moran's I is 0.341 and the $p$-value is 0.005 (<0.01). This is an indication of the strong spatial autocorrelation. Therefore, clustering robustness standard error is used to alleviate heteroscedastic and spatial autocorrelation interference. It is worth noting that, regardless of whether there is autocorrelation or not, in order to solve heterogeneity, the regression model in this article uses clustering robustness standard error, which is also the method to eliminate the interference of spatial autocorrelation;

4. Normality. On the one hand, both explained variables and the core explanatory variable are binary, and the method for estimating mediating effects is based on Logit model, which cannot satisfy the normality; on the other hand, the sample size of this study has reached 17,310. In the case of large samples, no normality test is required.

5.　Linearity. Based on Linktest, the regression coefficient of _hatsq is statistically insignificant (Coef. = 0.769; *p* = 0.411 (>0.1)). It means that the model specification is reasonable. Moreover, we replace the Logit model and test it with the Probit model and find that the results are still robust (Table A1);

6.　Endogeneity. To adequately control for the potential endogeneity issues associated with land registration variable, instrumental variables are incorporated into empirical analysis. The Instrumental Variables (IVs) of this article are defined as concern and investment-choice. 2SLS, LIML, IV-probit and IV-probit (two-stage) [79,80] for endogeneity testing are employed, and the specific model is as follows:

$$LI_i^* = \acute{\omega}_1 LRE_i + \acute{\omega}_2 IV_i + \acute{o}Z_1 + \rho_i \tag{12}$$

In Model 12, $LT_i^*$ represents the latent variable of the land transfer of the respondent i; $LRE_i$ represents whether respondent i has officially registered the land contracting and management rights; $IV_i$ represents the concern and investment choice of the respondent i; Other control variables represented by Zi include gender, health, agri-member, impoverishment, cadre, income, land-acreage, quality, and age; $\acute{\omega}_1$ and $\acute{\omega}_2$ are parameter estimates; $\acute{o}$ is the vector of parameter estimates; $\rho_i$ is the stochastic disturbance term.

## 4. Results

### 4.1. Descriptive Results

The descriptive statistics of the variables and the relationship between households' land registration and land transfer in this article are shown in Table 3, Figures 5 and 6. As shown in Figure 5, although the 28 provinces (autonomous regions and municipalities directly under the Central Government) have different proportions of land registration, they all exceed 20% (excluding Henan). Nationally, 46% of the farmers have completed their land registration, especially in 14 provinces (including Liaoning, Jilin, Heilongji, Jiangsu, Shandong, Hubei, Hunan, Hainan, Chongqing, Sichuan, Guizhou, Yunnan, Gansu, and Qinghai), where the proportion of land registration exceeds the overall average level. Figure 6 shows that households with land transfer account for 25% of the total. Among them, 42% have land transfer-out; 56% have land transfer-in; and 2% have both land transfer-out and transfer-in. According to the questionnaire, the main reasons for land transfer-out are "household not engaged in agricultural production"; the main reasons for land transfer-in are "meet our own needs" and "expand scale of agricultural production". Furthermore, households with land registration account for 46%. Among them, 12% have land transfer-out; 17% have land transfer-in; and 1% have both land transfer-out and transfer-in; while 70% have no land transfer. About 5.22% of the respondents hold the view that land registration isn't beneficial to farmers, and the main reasons are as follows: (1.) The rule of not increasing or not reducing by population is unfair to some farmers; (2.) farmers' awareness of safeguarding land rights is enhanced, so the land transfer progresses slowly; and (3.) land registration is averse to land consolidation.

Through the above descriptive statistics, it can be found that the proportion of households with land transfer is not high, and nearly half of them have officially registered the land contracting and management rights. The main reason why farmers transfer their land out is the migration of family labor to non-agricultural occupations. Moreover, the main reason why individual farmers or organizations transfer land in is to obtain higher agricultural returns. The exact quantitative relationship between land registration and land transfer and whether there is a logical relationship will be further discussed later.

### 4.2. Empirical Results

4.2.1. The Impacts of the Land Registration Program on Land Transfer

Table 4 examines H1, namely, whether the land registration program increases land transfer, including transfer-out and transfer-in. The models (1–3) in Table 4 show that under the control of other conditions, the regression coefficients of land registration are all

significantly positive at the 5% level. In addition, specifically, the chances of land transfer, land transfer-out, and land transfer-in with land registration increase by 24.7%, 37.1%, and 18.1%, respectively, compared with those without land registration. This indicates that the land registration program has a positive impact on land transfer, including transfer-out and transfer-in. Namely, on the one hand, under the premise that the land registration program increases the security perception of the land supply party, farmers who follow the discipline of maximizing profits are more inclined to transfer their land out. On the other hand, by fixing and clearly defining land contracting and management rights, the land registration program can enhance the dynamics of the land transaction market, promote the expectation of the availability of agricultural investment income, and then increase the possibility of land transfer-in [11]. In summary, H1 has been empirically verified.

**Table 3.** Descriptive statistics.

| Variables | Total | | Households with Land Registration | | Households without Land Registration | |
|---|---|---|---|---|---|---|
| | **Mean** | **SD** | **Mean** | **SD** | **Mean** | **SD** |
| Land transfer | 0.253 | 0.435 | 0.294 | 0.456 | 0.265 | 0.441 |
| Land transfer-out | 0.126 | 0.332 | 0.132 | 0.338 | 0.118 | 0.322 |
| Land transfer-in | 0.150 | 0.357 | 0.175 | 0.380 | 0.157 | 0.364 |
| Land registration | 0.459 | 0.498 | 1.000 | 0.000 | 0.000 | 0.000 |
| Security | 0.897 | 0.304 | 0.919 | 0.273 | 0.881 | 0.324 |
| Job-number | 1.040 | 0.200 | 1.041 | 0.203 | 1.044 | 0.207 |
| Agri-time | 7.334 | 3.696 | 7.493 | 3.663 | 7.139 | 3.694 |
| Crop-yield | 2.921 | 3.798 | 3.448 | 3.882 | 3.103 | 3.866 |
| Crop-value | 3.868 | 4.517 | 4.521 | 4.592 | 4.147 | 4.543 |
| Concern | 1.878 | 1.051 | 1.970 | 1.102 | 1.848 | 1.034 |
| Investment-choice | 4.205 | 1.142 | 4.177 | 1.154 | 4.239 | 1.142 |
| Gender | 0.521 | 0.500 | 0.522 | 0.500 | 0.522 | 0.500 |
| Health | 3.331 | 1.034 | 3.278 | 1.038 | 3.366 | 1.035 |
| Agri-member | 2.060 | 0.914 | 2.067 | 0.901 | 2.066 | 0.922 |
| Impoverishment | 0.161 | 0.367 | 0.167 | 0.373 | 0.148 | 0.356 |
| Cadre | 0.067 | 0.250 | 0.075 | 0.263 | 0.062 | 0.242 |
| Income | 7.486 | 3.922 | 7.308 | 4.051 | 7.615 | 3.797 |
| Land-acreage | 5.067 | 14.723 | 4.946 | 12.929 | 5.178 | 16.094 |
| Quality | 3.286 | 0.985 | 3.305 | 0.989 | 3.275 | 0.982 |
| Age | 39.454 | 21.243 | 39.999 | 21.158 | 38.750 | 21.095 |

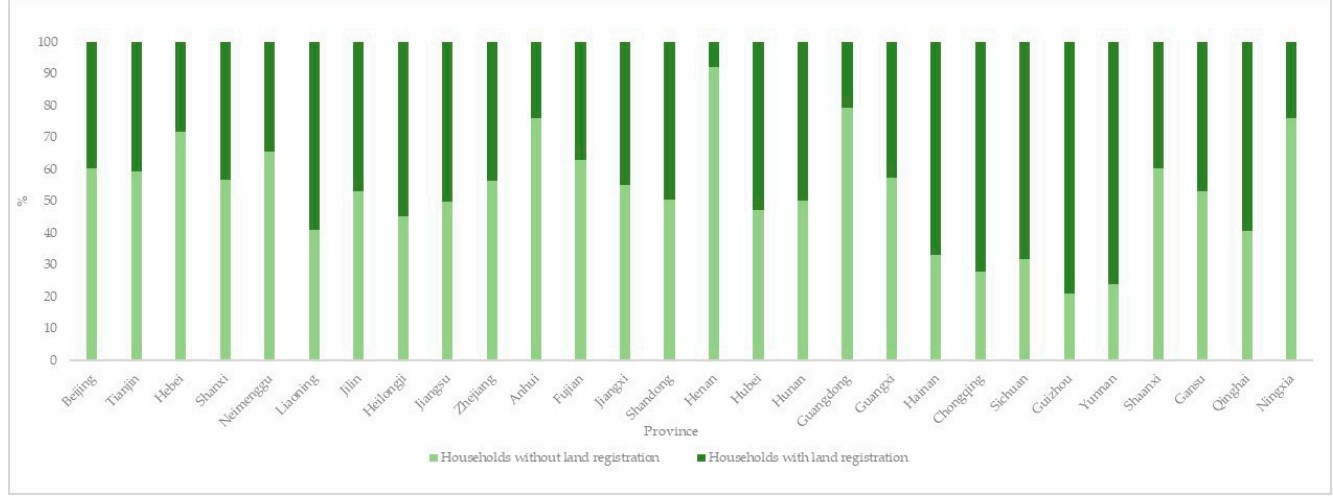

**Figure 5.** Land registration proportion in various provinces in rural China.

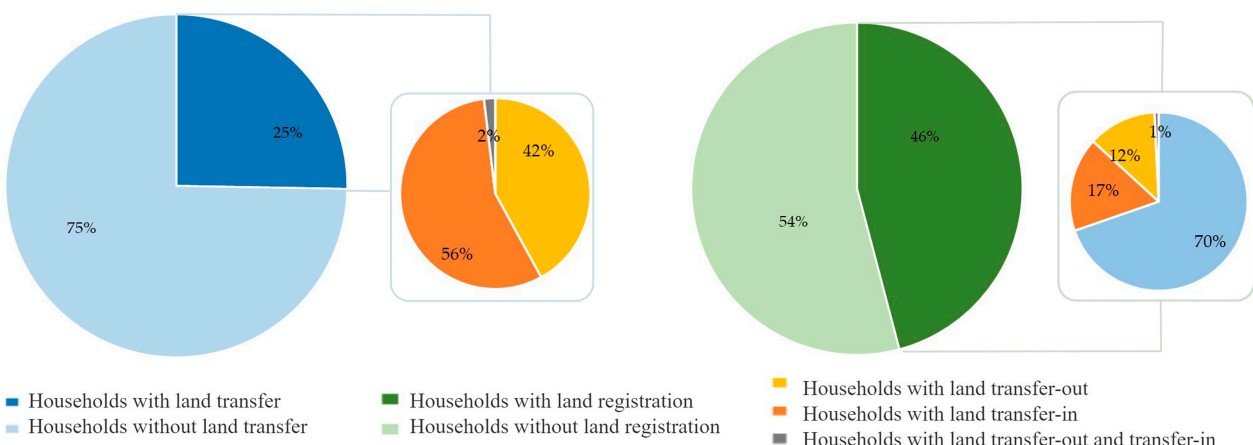

**Figure 6.** The relationship between households' land registration and land transfer in rural China.

**Table 4.** Estimation results of the Logit model of the impacts of the land registration program on land transfer.

| Variables | Model (1) | Model (2) | Model (3) |
|---|---|---|---|
| | **Land Transfer** | **Land Transfer-Out** | **Land Transfer-In** |
| Land registration | 0.221 *** | 0.316 ** | 0.167 ** |
| | [1.247] | [1.371] | [1.181] |
| | (0.079) | (0.126) | (0.085) |
| Gender | −0.030 | 0.039 | −0.055 ** |
| | [0.971] | [1.040] | [0.947] |
| | (0.022) | (0.039) | (0.024) |
| Health | 0.023 | 0.092 * | 0.002 |
| | [1.024] | [1.096] | [1.002] |
| | (0.027) | (0.052) | (0.030) |
| Agri-member | 0.005 | −0.228 ** | 0.081 ** |
| | [1.005] | [0.797] | [1.084] |
| | (0.040) | (0.099) | (0.040) |
| Impoverishment | −0.038 | −0.177 | −0.003 |
| | [0.962] | [0.838] | [0.997] |
| | (0.091) | (0.170) | (0.102) |
| Cadre | −0.057 | −0.038 | −0.051 |
| | [0.944] | [0.962] | [0.951] |
| | (0.124) | (0.227) | (0.138) |
| Income | 0.042 *** | −0.024 * | 0.062 *** |
| | [1.042] | [0.976] | [1.064] |
| | (0.009) | (0.014) | (0.011) |
| Land-acreage | 0.003 | −0.002 | 0.004* |
| | [1.003] | [0.998] | [1.004] |
| | (0.002) | (0.006) | (0.002) |
| Quality | 0.075 ** | 0.168 *** | 0.039 |
| | [1.078] | [1.183] | [1.040] |
| | (0.034) | (0.060) | (0.038) |
| Age | −0.002 | 0.007 *** | −0.005 *** |
| | [0.998] | [1.007] | [0.995] |
| | (0.001) | (0.002) | (0.002) |
| Cons. | −1.712 *** | −3.409 *** | −2.009 *** |
| | [0.181] | [0.033] | [0.134] |
| | (0.203) | (0.357) | (0.241) |
| Wald chi2 (10) | 53.58 | 36.29 | 77.29 |
| Pseudo R² | 0.008 | 0.015 | 0.013 |
| Obs. | 17,310 | 17,297 | 17,303 |

Note: Cluster Robust Standard Errors (CRSEs) in parentheses; OR in square brackets; * $p < 0.1$, ** $p < 0.05$, *** $p < 0.01$.

### 4.2.2. Mediating Effects of Livelihood Security and Agricultural Management Incentives

1. Suppression Effects: Livelihood Security

Models (1–5) in Table 5 show the estimation results of the suppression effects of livelihood security. This article uses job-number to measure livelihood security, and the results are shown in Models (1–3) of Table 5. Model (1) shows that under the premise of adding control variables, the probability ratio of land registration to land transfer-out is 1.371, and significantly positive at the 5% level. Furthermore, Model (2) shows that at the 5% level, the land registration significantly reduces job-numbers, and improves the security of farmers. The results in Model (3) show that at the 1% level, job number and land registration have significant impacts on land transfer-out, and the probability ratio of land registration has risen to 1.405. Furthermore, it should be noted that the regression coefficient $\alpha_2$ of land registration in Model (3) is 0.340, and the signs of $\alpha_1 \cdot \alpha_3$ ($-0.175 \times 0.385$) and $\alpha_2$ (0.340) are opposite. This means that for the land supply party, it is likely livelihood security plays a suppression effect and reduces the promoting impact of the land registration program on land transfer-out.

In order to increase the reliability of the conclusion, this article chooses security to measure livelihood security, and the results in Models (4–5) of Table 5 are consistent with the above conclusion. In summary, H2a is sufficiently verified.

2. Mediation Effects: Agricultural Management Incentives

Models (6–12) in Table 5 show the estimation results of the mediation effects of agricultural management incentives. This article uses agri-time to measure the agricultural management incentives, and the results are shown in Models (6–8) of Table 5. Model (6) shows that under the premise of adding control variables, the probability ratio of land registration to land transfer-in is 1.181, and significantly positive at the 5% level. Furthermore, Model (7) shows that at the 1% level, land registration significantly increases agricultural production and operation hours of individual farmers or agricultural economic organizations. The results in Model (8) show that agri-time and land registration have a significant impact on land transfer-in, and the probability ratio of land registration has dropped to 1.161, indicating that the land registration program has a significant impact on land transfer-in through extending agricultural business hours. Different from the suppression effects of livelihood security, the regression coefficient $\theta_2$ of land registration in Model (8) is 0.149, and the signs of $\theta_1 \cdot \theta_3$(0.371 × 0.023) and $\theta_2$(0.149) are consistent, indicating that agricultural management incentives mediate and increase the promoting impact of the land registration program on land transfer-in for the land demand party.

In order to increase the reliability of the conclusion, this article chooses to measure agricultural management incentives crop-yield and crop-value. The results in Models (9–12) of Table 5 show that the land registration program has a significant impact on the land transfer-in through crop-yield and crop-value, and the signs of $\theta_1 \cdot \theta_3$ and $\theta_2$ are also the same. The results are consistent with the above conclusion. In summary, H2b is sufficiently verified.

### 4.2.3. Robustness Test

The above has led to the conclusion that the land registration program could significantly improve land transfer. However, whether to officially register land contracting and management rights is an act of individual choice, and its impacts on land transfer will be restricted by household factors. To further verify the robustness of the above results, this article investigates the heterogeneity of impacts of the land registration program on land transfer from the perspectives of whether they are in the main grain producing area, agricultural mechanization, family members' roles, and family conditions (Figure 7).

**Table 5.** Estimation results of mediating effects of livelihood security and agricultural management incentives.

| Variables | Suppression Effects: Livelihood Security | | | | | | Mediation Effects: Agricultural Management Incentive | | | | | |
|---|---|---|---|---|---|---|---|---|---|---|---|---|
| | Model (1) | Model (2) | Model (3) | Model (4) | Model (5) | Model (6) | Model (7) | Model (8) | Model (9) | Model (10) | Model (11) | Model (12) |
| | Land Transfer-Out | Job-Number | Land Transfer-Out | Security | Land Transfer-Out | Land Transfer-In | Agri-Time | Land Transfer-In | Crop-Yield | Land Transfer-In | Crop-Value | Land Transfer-In |
| Land registration | 0.316 ** [1.371] (0.126) | −0.175 ** [0.840] (0.089) | 0.340 *** [1.405] (0.125) | 0.444 *** [1.559] (0.138) | 0.545 *** [1.725] (0.166) | 0.167 ** [1.181] (0.085) | 0.371 *** (0.137) | 0.149 * [1.161] (0.085) | 0.324 *** (0.125) | 0.147 * [1.159] (0.086) | 0.404 *** (0.102) | 0.152 * [1.165] (0.084) |
| Job-number | | | 0.385 *** [1.469] (0.146) | | | | | | | | | |
| Security | | | | | −0.448 * [0.639] (0.251) | | | | | | | |
| Agri-time | | | | | | | | 0.023 ** [1.023] (0.011) | | | | |
| Crop-yield | | | | | | | | | | 0.056 *** [1.057] (0.013) | | |
| Crop-value | | | | | | | | | | | | 0.085 *** [1.088] (0.011) |
| Control | Yes | Yes | Yes | Yes | Yes | Yes | Yes | Yes | Yes | Yes | Yes | Yes |
| Cons. | −3.409 *** [0.033] (0.357) | | −3.801 *** [0.022] (0.432) | 0.543 [1.721] (0.363) | −3.555 *** [0.029] (0.507) | −2.009 *** [0.134] (0.241) | 8.013 *** (0.348) | −2.184 *** [0.113] (0.252) | 3.945 *** (0.298) | −2.123 *** [0.120] (0.246) | 3.573 *** (0.301) | −2.261 *** [0.104] (0.251) |
| Pseudo $R^2$/$R^2$ | 0.015 | 0.057 | 0.015 | 0.024 | 0.022 | 0.013 | 0.013 | 0.014 | 0.034 | 0.015 | 0.089 | 0.023 |
| Obs. | 17,297 | 13,688 | 13,677 | 9554 | 9546 | 17,303 | 17,106 | 17,099 | 17,023 | 17,016 | 17,145 | 17,138 |

Note: Cluster Robust Standard Errors (CRSEs) in parentheses; OR in square brackets; * $p < 0.1$, ** $p < 0.05$, *** $p < 0.01$.

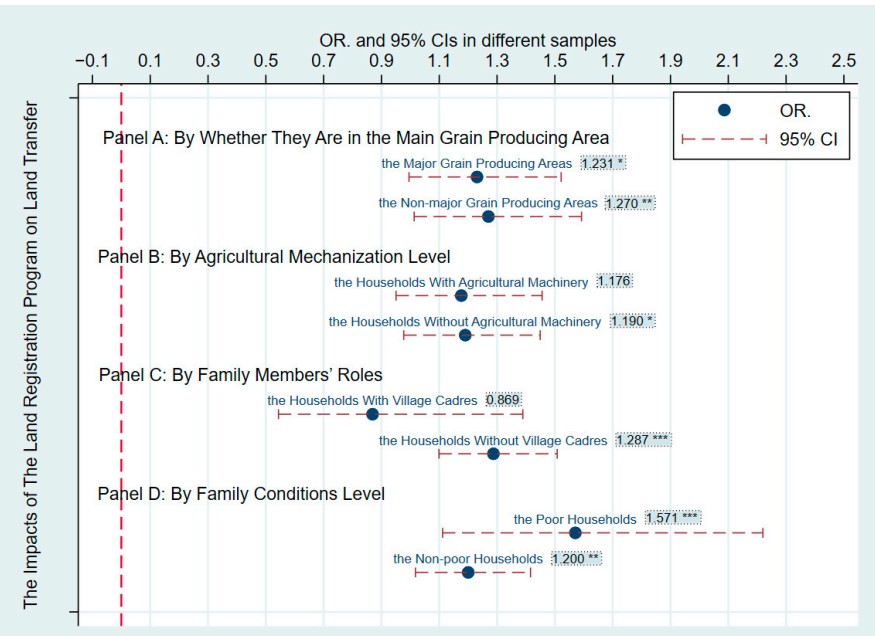

**Figure 7.** The impacts of the land registration program on land transfer by whether they are in the main grain producing area, agricultural mechanization level, family members' roles and family conditions level. The panel divides the whole sample into four sub-samples. Each point and the corresponding 90 percent confidence interval are based on a Logit model of Equation (3). The plotted points show the impacts of the land registration program on land transfer among the individuals in the corresponding sub-sample (* $p < 0.1$, ** $p < 0.05$, *** $p < 0.01$).

The main grain-producing areas in China are mainly in three major regions: the main grain-producing areas in Northeast China (Liaoning, Jilin, and Heilongjiang); the main grain-producing areas in Huanghuaihai (Hebei, Shandong and Henan); and the main grain-producing areas in the middle and lower reaches of the Yangtze River (Jiangsu, Anhui, Jiangxi, Hunan, Hubei, and Sichuan). First, in terms of whether they are in the main grain producing area, this article divides the whole sample into the major grain producing areas and the non-major grain producing areas. The results in Panel A of Figure 7 indicate that at the 5% level of significance, only land registration in non-major grain producing areas can significantly promote the land transfer of farmers. More specifically, their chances of land transfer after land registration have increased by 27.0%.

Second, regarding the level of agricultural mechanization, according to whether the agricultural machinery is used, this article divides the whole sample into households with agricultural machinery and households without agricultural machinery. The results in Panel B of Figure 7 reveal that whatever agricultural machinery is used, land registration can positively affect land transfer. However, at the 10% significance level, the impact of the land registration is significant only for farmers who do not use agricultural machinery. To be precise, their chances of land transfer after land registration have increased by 19.0%.

Thirdly, in terms of family members' roles, the results in Panel C of Figure 7 reveal that at the 1% level, the impact of the land registration is only significant in the samples of households without village cadres. More specifically, their chances of land transfer after land registration have increased by 28.7%.

Finally, in terms of family conditions level, the results in Panel D of Figure 7 indicate that regardless, of whether it is a poor family or not, the impact of the land registration on land transfer is significantly positive at the 5% level. Among them, the probability ratios of the sub-samples of poor and non-poor households are 1.571 and 1.200, respectively. More specifically, their chances of land transfer after land registration have increased by 57.1% and 20.0%. Relatively speaking, poor families have a higher chance of transferring their land after land registration.

4.2.4. Endogenous Variable Processing

In order to eliminate the possible selection bias, 2SLS, LIML, IV-probit, and IV-probit (two-stage) are used to conduct endogenous tests. In the first stage regressions of those models, the influence coefficients of the instrumental variables (concern and investment-choice) on land registration are significant, and the F values are greater than 10, showing that this article does not have the problem of weak instrumental variables [81]. Models (2–5) of Table 6 show that under other conditions remain unchanged, for respondents who are effectively interfered with by instrumental variables, there is a significant positive correlation between land registration and land transfer. So, the selected instrumental variables are reasonable. The endogenous treatment results once again verify the above research inferences; that is, on the basis of the effective instrumental variables, land registration significantly increases the incidence of land transfer.

**Table 6.** Instrumental variable estimation.

| Variables | Model (1) Logit | Model (2) 2SLS | Model (3) LIML | Model (4) IV-Probit | Model (5) IV-Probit (Two-Stage) |
|---|---|---|---|---|---|
| Land registration | 0.221 *** (0.079) | 0.592 *** (0.144) | 0.593 *** (0.145) | 1.380 *** (0.191) | 1.777 *** (0.433) |
| Control | Yes | Yes | Yes | Yes | Yes |
| Cons. | −1.712 *** (0.203) | −0.137 * (0.081) | −0.138 * (0.081) | −1.434 *** (0.073) | −1.848 *** (0.244) |
| Wald Chi2 (10) | 53.58 | 88.42 | 88.27 | 280.37 | 90.82 |
| Obs. | 17,310 | 15,188 | 15,188 | 15,188 | 15,188 |

Note: Robust Standard Errors in parentheses; * $p < 0.1$, *** $p < 0.01$.

## 5. Discussion

This article focuses on the mechanism of the land registration program on land transfer, bridges and extends concepts from Scott's survival ethics theory and the rational peasant theory to construct a theoretical framework of "the land registration program-behavioral disciplines-land transfer out/in," and conducts an empirical analysis based on mediation models and CHFS2015 data.

In line with existing studies [35,36], this article finds that the land registration program plays a significant role in promoting land transfer. Most studies either theoretically analyze the impacts of land systems on land transfer from a macro level [21–23] or explain the impacts of land systems on the behavior of stakeholders in a certain area from a micro level [24,25]. However, very few to date examine and test the differences in attitudes, subjective norms, and perceived behavior between the land supply and demand parties. In considering behavioral disciplines and cooperation of stakeholders in the process of land transfer, the interesting findings are that promotion of land value is inherent to rural households through weighing non-market value against market value. Both stakeholders' willingness to transfer land in/out and the occurrence of actual transfer behavior are rational decision-making processes affected by various socioeconomic factors. Additionally, the land supply and demand parties positively interact to realize effective communication and overcome the cooperative dilemma. The reasons might be that the land registration program reduces the asymmetry of land market information and transaction costs, standardizes land transaction behavior, and reduces land disputes. The primary objective of this article is to analyze the mechanism of the land registration program on land transfer in rural China from the perspectives of livelihood security and agricultural management incentives. Compared with existing studies, the possible contributions of this article are threefold:

1.   The article elaborates on the mechanism of the land registration program on land transfer, which is complementary to the existing literature in terms of research perspective and research content;

2. This article constructs a theoretical framework of "the land registration program-behavioral disciplines-land transfer out/in" and uses the mediation models to reveal and verify the logical relationship between the land registration program and land transfer out/in, which provides a novel perspective for a deep understanding of the intricate driving factors behind the decrease in the land transfer growth rate in rural China;

3. This article is instructive to re-examine the real needs of the peasant and provides policy implications to improve the rural social security system and the supporting system for land registration programs.

In addition, it should be pointed out that this article is limited to using CHFS 2015 data to analyze the impacts of the domestic land registration program on land transfer and its mechanism. We hope to verify and deepen the research using the latest data when the work of land registration is completed and the rural social security system is continuously improved in the future.

## 6. Conclusions and Implications

To free agricultural development from the constraint of arable land abandonment and extensive management, promoting the appropriate scale of land transfer has played an extremely important role in the rural revitalization in China. Over the years, the land registration program has stabilized the land property rights and mobilized farmers' enthusiasm for land transfer to a certain extent.

Facing the fact that the decrease of the land transfer growth rate in rural China has been more serious, this study testifies that the land registration program is beneficial to land transfer (including land transfer-out and transfer-in). In addition, agricultural management incentives promote the positive impact of the land registration program on land transfer-in. While security concerns limit the positive impact of the land registration program on land transfer-out. Therefore, Chinese society should recognize the fact that the rural social security system and land have gradually become the stabilizers and safety guarantee of rural society. Although farmers' dependence on their land has relatively decreased, it doesn't mean that the social security value contained in the land has decreased its attractiveness. In other words, livelihood security promotion still plays a key role in restricting the behavior of farmers.

In the future, more research should be done to consider the effects of the new-type rural social security system. In particular, rural social endowment insurance, rural cooperative medical care, and rural social assistance have benefited the majority of rural residents. The impact of those livelihood projects on farmers' land transfer behavior is a new area worthy of further exploration.

Based on the above discussion and conclusions, this article has two policy implications:

1. Suggesting the rural social security system should be optimized. Farmers' behavior choices are directly influenced by both the difficult situation of living without land and the situation of worrying after living without land. This reflects farmers' self-protection under the absence or imperfection of the rural social security system. Therefore, the government should actively respond to the changes in the rural social structure and rural residents' needs for a better life, promote the development of rural social security undertakings to a high-quality stage; and fully enhance farmers' survival security so that they can migrate to the secondary and tertiary industries and transfer their land out [34,36];

2. Suggesting the government should actively carry out the linkage reform of the supporting systems related to land registration programs, such as establishing a standardized land transaction market, promoting the land acquisition and compensation system, improving agriculture subsidy programs, promoting the efficient allocation of productive factors and resources in a multi-pronged manner, and laying a solid foundation for the sustainable and stable growth of farmers' income.

**Author Contributions:** Conceptualization, L.X. and S.C.; Data curation, L.X.; Formal analysis, L.X. and S.T.; Funding acquisition, S.C.; Methodology, L.X.; Software, L.X.; Supervision, S.C.; Writing—original draft, L.X. and S.T.; Writing—review and editing, L.X., S.T. and S.C. All authors have read and agreed to the published version of the manuscript.

**Funding:** This research was funded by the China Three Gorges Renewables (Group) Co., Ltd. (CTGR), grant number 1203/250000470; Division of Standardization, Administration for Market Regulation of Hubei Province, grant number 1203/250000471; and the Department of Education of Anhui Province, grant number SK2019A0309.

**Institutional Review Board Statement:** Not applicable.

**Informed Consent Statement:** Not applicable.

**Data Availability Statement:** Datasets are distributable only by the Survey and Research Center for China Household Finance of Southwestern University of Finance and Economics. They are available in the public domain on China Household Finance Survey (CHFS) website: https://chfs.swufe.edu.cn (accessed on 2 May 2022) and are also available on request from the corresponding author.

**Acknowledgments:** We would like to thank the Survey and Research Center for China Household Finance of Southwestern University of Finance and Economics, the China Three Gorges Renewables (Group) Co., Ltd. (CTGR), and all respondents for their contributions.

**Conflicts of Interest:** The authors declare no conflict of interest.

## Appendix A

In order to avoid disruption, the flow of the main text, Table A1, is placed in Appendix A. We replace the Logit model and test it with the Probit model. The results of Table A1 show a significant positive relationship between land registration and land transfer, which is consistent with the above conclusion.

**Table A1.** Estimation results of Logit and Probit models of the impacts of the land registration program on land transfer.

| Variables | Model (1) | Model (2) | Model (3) | Model (4) | Model (5) | Model (6) |
|---|---|---|---|---|---|---|
| | Probit | Logit | Probit | Logit | Probit | Logit |
| | Land Transfer | | Land Transfer-In | | Land Transfer-Out | |
| Land registration | 0.132 *** | 0.221 *** | 0.097 ** | 0.167 ** | 0.149 ** | 0.316 ** |
| | (0.047) | (0.079) | (0.048) | (0.085) | (0.060) | (0.126) |
| Gender | −0.017 | −0.030 | −0.029 ** | −0.055 ** | 0.019 | 0.039 |
| | (0.013) | (0.022) | (0.014) | (0.024) | (0.019) | (0.039) |
| Health | 0.014 | 0.023 | 0.000 | 0.002 | 0.043 * | 0.092 * |
| | (0.016) | (0.027) | (0.017) | (0.030) | (0.025) | (0.052) |
| Agri member | 0.002 | 0.005 | 0.047 ** | 0.081 ** | −0.101 ** | −0.228 ** |
| | (0.024) | (0.040) | (0.024) | (0.040) | (0.043) | (0.099) |
| Impoverishment | −0.025 | −0.038 | −0.004 | −0.003 | −0.086 | −0.177 |
| | (0.054) | (0.091) | (0.058) | (0.102) | (0.080) | (0.170) |
| Cadre | −0.033 | −0.057 | −0.029 | −0.051 | −0.031 | −0.038 |
| | (0.074) | (0.124) | (0.079) | (0.138) | (0.106) | (0.227) |
| Income | 0.024 *** | 0.042 *** | 0.034 *** | 0.062 *** | −0.012 * | −0.024 * |
| | (0.005) | (0.009) | (0.006) | (0.011) | (0.007) | (0.014) |
| Land acreage | 0.002 * | 0.003 | 0.003 ** | 0.004 * | −0.001 | −0.002 |
| | (0.001) | (0.002) | (0.001) | (0.002) | (0.002) | (0.006) |
| Quality | 0.044 ** | 0.075 ** | 0.022 | 0.039 | 0.082 *** | 0.168 *** |
| | (0.020) | (0.034) | (0.022) | (0.038) | (0.029) | (0.060) |
| Age | −0.001 | −0.002 | −0.003 *** | −0.005 *** | 0.003 *** | 0.007 *** |
| | (0.001) | (0.001) | (0.001) | (0.002) | (0.001) | (0.002) |
| Cons. | −1.032 *** | −1.712 *** | −1.185 *** | −2.009 *** | −1.881 *** | −3.409 *** |
| | (0.119) | (0.203) | (0.135) | (0.241) | (0.169) | (0.357) |
| Wald chi2 (10) | 53.65 | 53.58 | 78.82 | 77.29 | 36.19 | 36.29 |
| Pseudo R$^2$ | 0.008 | 0.008 | 0.013 | 0.013 | 0.015 | 0.015 |
| Obs. | 17,310 | 17,310 | 17,303 | 17,303 | 17,297 | 17,297 |

Note: Cluster Robust Standard Errors (CRSEs) in parentheses; * $p < 0.1$, ** $p < 0.05$, *** $p < 0.01$.

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
