# Peer review of "The Mechanism of Land Registration Program on Land Transfer in Rural China: Considering the Effects of Livelihood Security and Agricultural Management Incentives"

_land, doi:10.3390/land11081347_

Round 1

Reviewer 1 Report

I suggest the authors test the regressions for multicollinearity, autocorrelation (including spatial), heteroscedasticity, endogeneity, normality, linearity, adequacy of the model.

In addition, it is important to explain the concepts used in the abstract. 

Reviewer 2 Report

         The issue of appropriate scale management of land is related to food security and has always been a hot topic in academia and politics. The author focuses on the influence of land right confirmation on land transfer and tries to evaluate its policy effect. This is interesting work, but the results do not seem to support the authors. Specific comments are as follows:

         (1) The problem of data time. The author used the CHFS data of 2015 (the actual data collected was the end of 2014). This dataset may be slightly insufficient for analyzing the impact of land right confirmation on land transfer. The reasons are as follows: Land right confirmation was only piloted in some places in 2013 or 2014, and has not yet been implemented in many places. It is not appropriate to evaluate the effect of this policy based on data from this time point. At the same time, China completed the confirmation of land rights around 2019. In fact, in my opinion, to systematically evaluate the impact of land right confirmation projects on land transfer, dynamic panel data should be used to capture dynamic changes, rather than static time-point data at the initial stage of project implementation.

         (2) The problem of setting the scene. In fact, why does land right confirmation in some places have an obvious effect on land transfer, while in some places it is not obvious. This is actually related to the resource endowment of the region. Very simple truth, for the northeast black land, the vast land sparsely populated, convenient mechanization, the implementation of land right confirmation project can further promote the appropriate scale of land management. Farmers can increase their income through moderate scale management and socialized services. However, at the same time, it is noted that in many mountainous areas (such as Sichuan Province and Chongqing Municipality), the per capita arable land is less than 1 mu, and the average household is less than 10 mu, so it is difficult to earn a few cents without eating or drinking the land to grow food. Under such resource endowment, it is normal that the incentive effect of land right confirmation has no effect. Therefore, I think the author does not grasp the key of the problem. Perhaps the sample area of the study will be divided according to the terrain, according to the major grain producing areas and non-major grain producing areas, and according to the level of mechanization, and further heterogeneity analysis will be more enlightening.

         (3) Some important things the authors conspicuously missed, such as:

The impact of farmland property rights security on the farmland investment in rural China.

The impact of the land certificated program on the farmland rental market in rural China.

Rural-urban migration and its effect on land transfer in rural China.

Round 2

Reviewer 1 Report

I suggest the authors rethink and improve the explanations given for the potential statistical problems. For example, it would be important to present the results for the several tests used (Moran I,...). 

Reviewer 2 Report

I have no other comments, thank you.

Author Response

Reviewer 2's second-round comment is <accept>

This manuscript is a resubmission of an earlier submission. The following is a list of the peer review reports and author responses from that submission.

Round 1

Reviewer 1 Report

About the paper "Land Registration Program and Land Transfer Behavior in Rural China: Considering the Effects of Survival Security and Agricultural Management Incentives" I have the following comments:

The paper is very hard to follow and this begins in the title and in the abstract. This reflects, in fact, a confuse definition of the objectives and methodologies.

For example, in the abstract, the authors referred "Existing studies explore how land registration program affects land transfer, but little consideration has been given to the interaction and transfer behavioral disciplines of stakeholders (including land supply and demand parties).". From here it seems that the land registration impacts on land transfer is already explored by other studies. But, in the next sentence for the objectives, the authors written "This article aims at a more in-depth investigation of the impact of land registration program on land transfer behavior in rural China...". I suggest the authors clarify the title and the abstract with: motivations and gaps in the literature that justify the paper, present the objectives clearly, methodologies, novelties and main insights.

The methodologies considered ("Logit and Biprobit models") was not properly justified and supported by scientific literature. In addition, survival analysis has proper statistical appoaches not described and presented in the manuscript.

On the other hand, the results of the regressions were presented without any concern (without the proper statistical tests) about multicollinearity, heteroscedasticity, adequacy of the models, autocorrelation (time-series and spatial), linearity, endogeneity, normality, ....

The conclusions section, should be considered to present the main insights, practical implications, policy recommendations and suggestions for future research.

Reviewer 2 Report

Surely, exploring the impact of land registration program on land transfer behavior (transfer-out and transfer-in) in rural China is a very important topic and valuable to be studied. Overall, this manuscript was well written and organized and logic, and authors have done much work on it. However, due to the data being too old, I doubt it implication for current or future land reform in China. This can also be confirmed by  construction of the modern farmland property right system after 2015 (L124-139 as described in the current version). In terms of '...in 2020 once again clarified the rights and interests of land contracting and management rights holders, and eased farmers' concerns about losing land', you can know that 'easing farmer' concerns about losing land' will significantly affect land transfer behavior of land supply or transfer-out. 

In this case, I suggest that you should do yourself investigation as results of this study are not time-based.  If you could do that, I am sure that issues your concerned will be well addressed. 

Reviewer 3 Report

First of all, there are essentially two levels of data. The first level is the data of households’ land transfer or land renting. Another is the data of land registration. The implementation of land registration is organized at rural community-level or township-level. That is to say, the data is nested, namely, households are nested in community, township or county. Therefore, the question is whether the model specification is right or correct.

Please give a more detailed background and Please explain the policy of farmland or agricultural land registration and its changes over past 40 years. Was there no agricultural land registration in 1980s, 1990s and 2000s?

It is not clear for what is the cooperation between transfer-in and transfer-out. What is the definition?

I think it is necessary to give an in-text citation for the sentence in line 205-206.

Line 39 to 43: Please give the proportion of land rented-in or rented-out.